# Stigma towards mental illness and help-seeking behaviors among adult and child psychiatrists in Hungary: A cross-sectional study

Dorottya Őri[1,2]*, Péter Szocsics[3], Tamás Molnár[4], Fanni Virág Ralovich[3], Zsolt Huszár[3], Ágnes Bene[5], Sándor Rózsa[6], Zsuzsa Győrffy[2], György Purebl[2]

1 Department of Mental Health, Heim Pál National Pediatric Institute, Budapest, Hungary, 2 Institute of Behavioural Sciences, Semmelweis University, Budapest, Hungary, 3 Department of Psychiatry and Psychotherapy, Semmelweis University, Budapest, Hungary, 4 University of Pécs Medical School, County Hospital Győr, Petz Aladár Hospital, Győr, Hungary, 5 Department of Psychiatry and Psychiatric Rehabilitation, Saint John Hospital, Budapest, Hungary, 6 Department of Psychiatry, Washington University School of Medicine, St. Louis, Missouri, United States of America

* oridorottya@gmail.com

**Citation:** Őri D, Szocsics P, Molnár T, Ralovich FV, Huszár Z, Bene Á, et al. (2022) Stigma towards mental illness and help-seeking behaviors among adult and child psychiatrists in Hungary: A cross-sectional study. PLoS ONE 17(6): e0269802. https://doi.org/10.1371/journal.pone.0269802

**Data Availability Statement:** Data are held in a public repository: DOI: https://doi.org/10.3886/E171541V1 URL: https://www.openicpsr.org/openicpsr/project/171541/version/V1/view.

# Abstract

## Objective

Stigma towards people with mental health problems is a growing issue across the world, to which healthcare providers might contribute. The aim of the present study was to explore psychiatrists' attitudes towards their patients and link them to psychosocial and professional factors.

## Methods

An online questionnaire was used to approach the in- and outpatient psychiatric services across Hungary. A total of 211 trainees and specialists in adult and child psychiatry participated in our study. Their overall stigmatizing attitudes were measured, with focus on attitude, disclosure and help-seeking, and social distance dimensions by using the self-report Opening Minds Stigma Scale for Health Care Providers (OMS-HC). Multiple linear regression analyses were performed to elucidate the dimensions of stigma and its association with sociodemographic, professional and personal traits.

## Results

Stigmatizing attitudes of close colleagues towards patients were statistically significant predictors of higher scores on the attitude [B = 0.235 (0.168–0.858), p = 0.004], the disclosure and help-seeking subscales [B = 0.169 (0.038–0.908), p = 0.033], and the total score of the OMS-HC [B = 0.191 (0.188–1.843), p = 0.016]. Psychiatrists who had already sought help for their own problems had lower scores on the disclosure and help-seeking subscale [B = 0.202 (0.248–1.925), p = 0.011]. The overall stigmatizing attitude was predicted by the openness to participate in case discussion, supervision or Balint groups [B = 0.166 (0.178–

**Funding:** The author(s) received no specific funding for this work.

**Competing interests:** The authors have declared that no competing interests exist.

5.886), p = 0.037] besides the more favorable attitudes of their psychiatrist colleagues [B = 0.191 (0.188–1.843), p = 0.016].

## Conclusions

The favorable attitudes of psychiatrists are associated with their own experiences with any kind of psychiatric condition, previous help-seeking behavior and the opportunity to work together with fellow psychiatrists, whose attitudes are less stigmatizing. The perception of fellow colleagues' attitudes towards patients and the openness to case discussion, supervision and Balint groups were the main two factors that affected the overall attitudes towards patients; therefore, these should be considered when tailoring anti-stigma interventions for psychiatrists.

## Introduction

Stigma is an implicit or explicit social rejection originating from putative or actual negative characteristics of the individual or groups of individuals [1]. Stigmatization leads to a "spoiled identity" that negatively influences the life of the stigmatized person. People with mental health problems experience stigmatizing attitudes from the public and healthcare professionals as well [2], leading to the internalization of these beliefs [3]. This phenomenon alters the quality of life of the affected and depresses their life in the workplace, private and public environment as well as healthcare services [4]. It should be highlighted that people with mental health disorders have a shorter life expectancy that might be influenced by the stigma of help-seeking and the lack of treatment as a consequence [5]. Therefore, reducing discrimination and mental health-related stigmatization is highly beneficial for the affected people in many ways that might improve their overall well-being. Being experts in this field, mental health professionals are in the right position to reduce stigma via their clinical work and through the education of the public. However, it must be admitted that mental health related stigma appears in mental health care as well. In order to implement successful stigma reduction methods, it is crucial to gain a better understanding of factors that could influence these attitudes and to identify those that are protective against discrimination.

Several studies have examined the attitudes towards people with mental illnesses in the public and the healthcare sector [6–8]. Most of the results show that the public has more negative assumptions towards these people than healthcare workers. Negative attitudes might come from the lack of knowledge about mental disorders. Consequently, mental health professionals are the least stigmatizing group among healthcare workers. A study investigating students from different medical schools revealed that being in education programs in which mental health issues were prioritized was negatively correlated with stigmatizing attitudes [9]. However, in some cases, the direct and frequent contact with people with mental health problems has led to more stigmatization among mental health professionals and psychiatrists [10, 11]. These controversies might be explained by cultural reasons shown in a multicenter clinical trial involving various healthcare professionals from five different countries [12].

Only few investigations have been conducted to examine the stigmatizing attitudes of psychiatrists. There is conflicting evidence about psychiatrists holding stigmatizing views, to summarize the findings, psychiatrists seem to be less stigmatizing compared to other specialists [9, 13, 14]. A large Canadian study comprised of the population of 12 different anti-stigma programs for various healthcare professionals within the 15-year ongoing Opening Minds anti-

stigma initiative reached the same conclusion, when comparing psychiatrists' views to family physicians, rural physicians, anesthetists and surgeons [15]. Their results regarding psychiatrists are comparable with our findings, since they developed the Opening Minds Stigma Scale for Healthcare Providers (OMS-HC) that was used in this study to measure stigmatizing attitudes.

Most recently, Brazilian psychiatrists were grouped based on their attitudes towards people with schizophrenia into high, intermediate and low stigma categories. Surprisingly, half of the study population was categorized into the high stigma group. The most stigmatizing group spent significantly longer time at work since graduation, had higher anxiety-state scores and lower positive affect [16]. According to a recent meta-analysis, the demand for personal accomplishment and burnout are predictors of more stigmatizing attitudes among healthcare providers [17]. A Belgian study investigating both professionals and service users pointed out that the associative stigma of mental health care providers was related to more self-stigma and dissatisfaction of the clients, besides the increased symptoms of burnout and the less job satisfaction of mental health professionals [18]. Therefore, as prevention, mental healthcare workers should be provided with supportive opportunities such as skill-based interventions [19], intensive social contact intervention with persons with lived experience of mental illness [20], unconscious bias informed education [21], and mindfulness based interventions to reduce the stress [22]. The Balint group is a patient-centered approach that deepens the understanding of the doctor-patient relationship and improves communication skills, which is protective against burnout syndrome [23]. It is also considered a valuable tool to reduce stress in physicians, which can consequently improve attitudes and the care they provide to patients [24]. Active psychotherapeutic practice usually also goes hand in hand with case discussion groups, which provide the participants with new skills to manage patients and reduce anxiety and stress. These tools improve the clinicians' awareness and provide them with new skills, which are key elements of the more effective treatment and better outcomes. Lack of skills and training could cause healthcare providers to keep more distance from patients and feel less confident, resulting in negative attitudes and poorer treatment outcomes [25]. Accordingly, since higher levels of burnout and low personal accomplishment were associated with more negative attitudes towards patients among healthcare workers, their support and education might play an essential role in increasing their confidence and mitigating their stigmatizing attitudes [26]. The results of a multicenter study, which included a Hungarian sample as well, revealed that people with mental health disorders experience stigma in private life, at work and in health care settings. The stigmatization seemed to be higher in post-communist countries as compared with others [4]. Nonetheless, the stigmatizing attitudes of mental health professionals and psychiatrists had never been studied in Hungary.

We aimed to investigate the stigma construct in-depth by examining associations with sociodemographic and professional factors, personal experiences and mental health problems in a sample of specialists and trainees in adult, child and adolescent psychiatrists in Hungary.

## Materials and methods

### Study overview and participants

This was a cross-sectional, observational anonymous online survey study conducted to measure the stigmatizing attitudes of trainees and specialists in adult, child and adolescent psychiatry in Hungary. Altogether an estimated 972 professionals in psychiatry, from which the estimated maximum of 635 work in the public health sector (trainees in general adult psychiatry: 140, trainees in child and adolescent psychiatry: 47, specialists in general adult psychiatry: 700 –of whom 403 work in the public health sector -, specialists in child and adolescent

psychiatry: 85) work in Hungary. Our research group contacted a total of 50 adult and 10 child psychiatric inpatient services together with 52 adult and 17 child outpatient services across Hungary, serving both urban and rural areas, via e-mail and telephone. The survey link was sent directly to the head of the unit, who was asked to forward it to the psychiatrist colleagues. To broaden the study population, the survey link was also shared through social media platforms. It was included in the newsletter of the Hungarian Association of Psychiatric Trainees and the Hungarian Psychiatric Association.

## Measurements

**Psychosocial and demographic data.** Ten direct questions on demographic information and twelve questions on the participants' mental health, their close family members, attitudes towards psychotherapy and case discussion groups were raised. The following details were gathered: age range, sex, work status, years of experience in psychiatry, type and place of the working institute, having any friend or family member with mental health problems, own experience with mental health problems, experience in psychotherapy, accessibility and demand for attending case discussion, supervision or Balint groups, and the stigmatizing attitudes of colleagues who work in the institute as the participant.

**Stigmatizing attitudes.** The OMS-HC is a widely used self-report questionnaire that contains statements describing feelings and opinions about people with mental health problems. Subjects indicate on a five-point Likert scale (1 = "strongly disagree" to 5 = "strongly agree") the extent to which each given statement characterizes them. Five items are reverse coded (1 = "strongly agree" to 5 = "strongly disagree"). The overall stigmatizing attitudes of the participants are described with the total score of the scale (a minimum of 14 and a maximum of 70 points). Besides the total score, three dimensions can be calculated by evaluating the three subscales of the questionnaire (Attitude, Disclosure and Help-seeking and Social distance). Higher scores reflect more stigmatizing attitudes.

The original 20-item version of the scale [27] was shortened to 15 items by Modgill [15] to gain a more stable factor solution. The scale structure has also been investigated in Italy, Chile and Singapore [28–30]. The Hungarian version of the OMS-HC has strong psychometric properties [31]. It shows a bifactor structure, in which the three dimensions of the stigma (Attitude, Disclosure and Help-seeking as well as and Social distance) have been confirmed, and it consists of 14 statements. Its test-retest reliability was found to be excellent for the total score and good for the three subscales, and a good concurrent validity was measured for the total score with the Mental Illness: Clinician's Attitudes-4 scale (intraclass correlation coefficient value was 0.77 [95% confidence interval ranged between 0.11 and 0.92]).

## Ethical considerations

The study was approved by the Regional and Institutional Committee of Science and Research Ethics of the Semmelweis University, Budapest, Hungary (approval number: SE-RKEB: 189/2019) and was conducted in accordance with the principles of the Declaration of Helsinki. Prior to the enrolment, all of the participants provided their informed consents via the online survey.

## Statistical approach

Demographic data are expressed as the sample size (n) with percentages (%). Since the data did not follow the normal distribution, the median and interquartile ranges were calculated (IQR) for continuous variables, and non-parametric tests were performed. For comparison of two groups, we used the Mann-Whitney U test with the Monte Carlo simulation method,

whereas we applied a non-parametric one-way ANOVA test for more than two groups. Stepwise multiple linear regression analyses were performed using the OMS-HC total and subscale scores as dependent variables and personal and professional factors as independent variables. Standardized Betas, p-values, 95% confidence intervals and explained variances ($r^2$) are listed as indicators. $R^2$ was calculated as an index of the percentage of the variation in the criterion (stigma scores) explained by the predictors (personal and professional factors). Statistical significance was reported at $p < 0.05$. The following softwares were used for the statistical analyses: IBM SPSS 25 (Apache Software Foundation, USA) and GraphPad Prism 8.0.1 (GraphPad Softwares Inc, USA).

## Results

A total of n = 238 professionals in psychiatry responded to our survey, the data of 211 subjects were analyzed who completed the entire survey. Table 1 shows the sociodemographic characteristics of the subjects. The majority of the participants were female (76%), with the overrepresentation of young colleagues between 24 and 35 years of age (54%). Most of them were adult psychiatrists (64%) and specialists (57%). In approximately two-thirds of the cases, they worked in inpatient services (66%), and their workplace was in the capital (62%).

**Table 1. Sociodemographic characteristics of the participants.**

| Variables | | n (%) |
|---|---|---|
| **Age group (years)** | 24–35 | 114 (54) |
| | 36–45 | 34 (16) |
| | 46–55 | 38 (18) |
| | 56–65 | 16 (8) |
| | 66–75 | 9 (4) |
| **Gender** | Male | 50 (24) |
| | Female | 161 (76) |
| **Professional group** | Trainee | 90 (43) |
| | Specialist | 121 (57) |
| **Location of workplace** | Capital | 131 (62) |
| | County capital | 61 (29) |
| | City | 10 (5) |
| | Town | 9 (4) |
| **Type of practice they work in the majority of their working hours** | Inpatient hospital | 139 (66) |
| | Outpatient service | 55 (26) |
| | Exclusively private practice | 9 (4) |
| | Daycare service | 5 (2) |
| | Other | 3 (2) |
| **Ever experienced negative discrimination for working as a psychiatrist** | Yes | 91(43) |
| | No | 119 (57) |
| **Having friends or family members with mental illness** | Yes | 124 (59) |
| | No | 77 (36) |
| | I do not know | 9 (4) |
| **Ever been medically treated for any psychiatric problems** | Yes | 41 (19) |
| | No | 169 (80) |
| **Ever been participating in psychotherapy for any reason** | Yes | 156 (74) |
| | No | 50 (24) |

Descriptive statistics were presented.

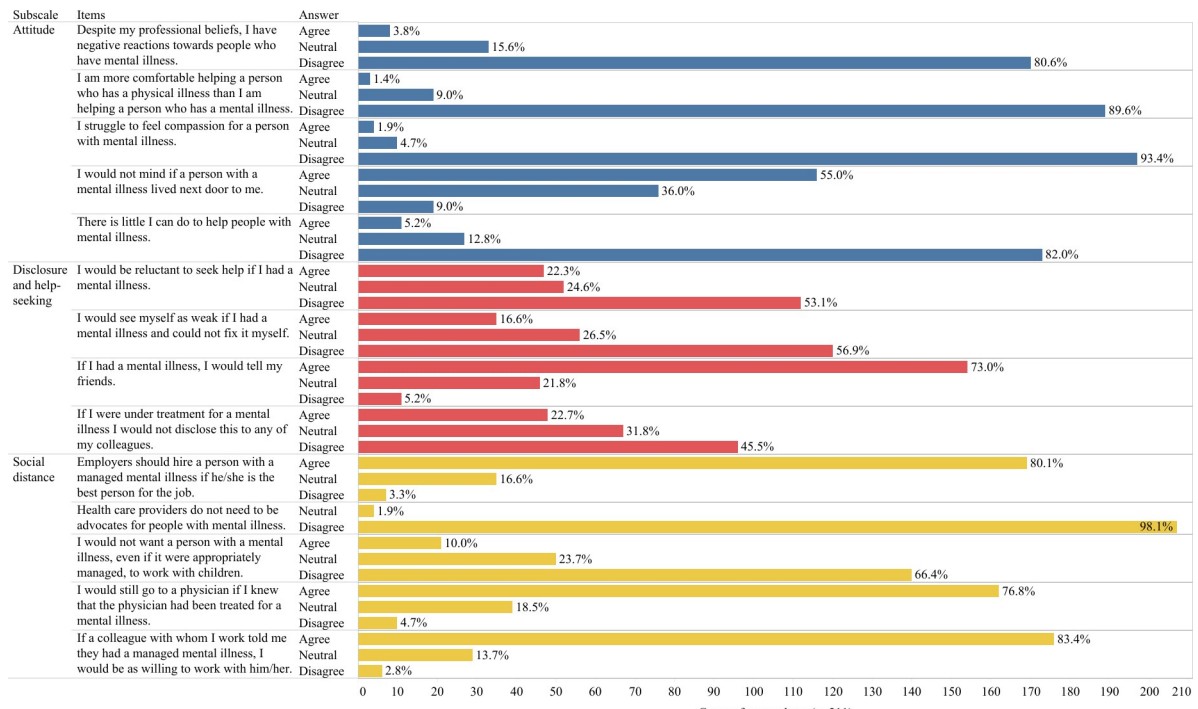

**Fig 1. Distribution of responses to items of the Opening Minds Stigma Scale for Health Care Providers (14 items).**

Fig 1 presents the distribution of responses to each OMS-HC item. On attitude and social distance subscales, the responses clearly indicated that the majority of the participants have positive attitudes towards people with mental health problems. As Fig 1 illustrates, the results concerning disclosure and help-seeking statements were mixed. One half of the subjects agreed, and the other half was neutral or disagreed with the following items: "I would be reluctant to seek help if I had a mental illness" and "If I were under treatment for a mental illness, I would not disclose this to any of my colleagues"; however, 73% of them would tell their friends if they had a mental illness.

The total score of the 14-item OMS-HC could range from 14 to 70, subscale scores for attitude, disclosure and help-seeking, and social distance vary between 5 and 25, 4 and 20, and 5 and 25, respectively. Table 2 shows that the overall stigmatizing attitudes of participants, which is represented by the total score (median: 29, IQR: 26–32), as well as their attitude (median: 10, IQR: 8–11) and social distance subscale (median: 10, IQR: 8–11) scores, were in the lowest third of the maximum reachable score. However, disclosure and help-seeking (median: 10, IQR: 8–12) were relatively higher since only four items belong to this subscale; therefore, the reachable maximum score is lower. There were no statistically significant differences by grouping respondents according to age ranges, sex, work status, years of experience in psychiatry, type and place of institute, and the mental health problem of any of their close friends or family members. Conversely, we found statistically significant differences between the subscale scores of participants categorized into different groups based on psychotherapeutic activity, seeking help for their own mental health problems, openness and having a possibility to participate in case discussion, supervision or Balint groups and attitudes of close psychiatrist colleagues towards patients.

To further investigate the relationship between the total and subscale scores of the OMS-HC and the characteristics of the participants, multiple regression analyses were

**Table 2. OMS-HC subscale scores and total scores of the participants.**

| | | n | (%) | Attitude Median | Attitude IQR | p | Disclosure and help-seeking Median | Disclosure and help-seeking IQR | p | Social distance Median | Social distance IQR | p | 14-item total Median | 14-item total IQR | p |
|---|---|---|---|---|---|---|---|---|---|---|---|---|---|---|---|
| Overall | | 211 | 100 | 10 | 8–11 | | 10 | 8–12 | | 10 | 8–11 | | 29 | 26–32 | |
| Professional group | Adult | 135 | 66 | 10 | 8–11 | 0.385 | 10 | 9–12 | 0.053 | 9 | 7–11 | 0.525 | 29 | 26–33 | 0.706 |
| | Child | 69 | 34 | 10 | 8–11 | | 9 | 8–11 | | 10 | 8–11 | | 29 | 26.5–32 | |
| Actively provides psychotherapy | Yes | 101 | 48 | 9 | 8–11 | **0.028** | 10 | 8–11 | 0.877 | 10 | 8–11 | 0.749 | 29 | 26–32 | 0.507 |
| | No | 110 | 52 | 10 | 8–11 | | 10 | 8–12 | | 9 | 8–11 | | 29 | 25.75–33 | |
| Ever sought help for own mental health problems | Yes | 98 | 46 | 10 | 8–11 | 0.416 | 9 | 7.75–12 | **0.042** | 9 | 7–11 | **0.037** | 28.5 | 24.75–32 | 0.107 |
| | No | 111 | 53 | 10 | 8–11 | | 10 | 9–12 | | 10 | 8–11 | | 29 | 26–33 | |
| Be open to participating in case discussion groups* | Yes | 184 | 87 | 10 | 8–11 | 0.182 | 10 | 8–11 | **0.012** | 9 | 7–11 | **0.016** | 29 | 25–32 | **0.004** |
| | No | 27 | 13 | 10 | 9–12 | | 11 | 9–13 | | 11 | 9–11 | | 32 | 28–35 | |
| Has a possibility to participate in case discussion groups* | Yes | 132 | 63 | 10 | 8–11 | 0.126 | 10 | 8–11 | 0.251 | 9 | 7–11 | 0.186 | 28 | 25.25–32 | **0.046** |
| | No | 79 | 37 | 10 | 8–12 | | 10 | 9–12 | | 10 | 8–11 | | 30 | 27–33 | |
| Close psychiatrist colleagues' stigmatizing attitudes towards their patients | not at all | 48 | 23 | 9 | 7–10 | **0.017** | 9 | 8–11 | **0.021** | 9.5 | 7–10.75 | 0.240 | 27 | 25–32 | **0.044** |
| | to small extent | 75 | 36 | 10 | 8–11 | | 9 | 8–11 | | 9 | 7–11 | | 29 | 24–32 | |
| | to some extent | 62 | 29 | 10 | 8–12 | | 10 | 9–12 | | 10 | 8–11 | | 30 | 27–34 | |
| | to great extent | 25 | 12 | 10 | 9–11 | | 11 | 10–12.5 | | 9 | 8–10 | | 31 | 27.5–33 | |

* Case discussion groups, supervision or Balint groups

Mann-Whitney U test with Monte Carlo Simulation and Non-parametric Analysis of variance was applied. The statistically significant results (p < 0.05) are in bold.

performed. Table 3 presents that most of the differences remained statistically significant in the prediction model; moreover, some other factors were revealed to be statistically significant predictors of stigma. Altogether, the tested variables could explain 10% of the attitude, 7% of the disclosure and help-seeking, 9% of the social distance subscale and 7% of the total stigma score. The results showed that the more stigmatizing attitudes of close colleagues towards patients were a statistically significant predictor of higher scores on the attitude [B = 0.235 (0.168–0.858), p = 0.004], the disclosure and help-seeking subscales [B = 0.169 (0.038–0.908), p = 0.033], and the total score of the OMS-HC [B = 0.191 (0.188–1.843), p = 0.016]. Active psychotherapeutic practice [B = 0.179 (0.099–1.425), p = 0.025] and the experience of negative discrimination for working as a psychiatrist [B = 0.163 (0.024–1.364), p = 0.042] predicted lower attitude scores. Psychiatrists who had ever sought help for their own problems had lower scores on the disclosure and help-seeking subscale [B = 0.202 (0.248–1.925), p = 0.011]. Being ever medically treated for any kind of mental health problems [B = 0.184 (0.185–2.063), p = 0.019] and the experience of negative discrimination for being a psychiatrist [B = 0.245 (0.439–1.931), p = 0.043] were predictors of lower social distance scores. The overall

**Table 3. Standardized beta coefficient estimates represent the analysis of OMS-HC total and subscales scores adjusted for professional and personal factors.**

|  | Beta | p-value | 95%CI | $r^2$ |
|---|---|---|---|---|
| **Attitude** | | | | |
| Close psychiatrist colleagues' stigmatizing attitudes towards their patients | 0.235 | 0.004 | 0.168–0.858 | 0.104 |
| Actively provides psychotherapy | 0.179 | 0.025 | 0.099–1.425 | |
| Has experienced negative discrimination for working as a psychiatrist | 0.163 | 0.042 | 0.024–1.364 | |
| **Disclosure and Help-seeking** | | | | |
| Ever sought help for own mental health problems | 0.202 | 0.011 | 0.248–1.925 | 0.070 |
| Close psychiatrist colleagues' stigmatizing attitudes towards their patients | 0.169 | 0.033 | 0.038–0.908 | |
| **Social distance** | | | | |
| Has experienced negative discrimination for working as a psychiatrist | 0.245 | 0.002 | 0.439–1.931 | 0.090 |
| Ever been medically treated for a mental health problem | 0.184 | 0.019 | 0.185–2.063 | |
| **14-item total** | | | | |
| Close psychiatrist colleagues' stigmatizing attitudes towards their patients | 0.191 | 0.016 | 0.188–1.843 | 0.067 |
| Be open to participating in case discussion groups* | 0.166 | 0.037 | 0.178–5.886 | |

* Case discussion groups, supervision or Balint groups.

Multiple linear regression analysis using stepwise method. 95% CI = 95% Confidence interval, $r^2$ = Rho square. Only the statistically significant results are presented ($p \leq 0.05$).

stigmatizing attitude was predicted by the openness to participate in case discussion, supervision or Balint groups [B = 0.166 (0.178–5.886), p = 0.037] besides the more favorable attitudes of their psychiatrist colleagues [B = 0.191 (0.188–1.843), p = 0.016].

## Discussion

Due to there being a scarcity of studies in the literature investigating stigma among mental healthcare providers (the majority being social workers, nurses, psychologists, etc.), the authors of the present study aimed to investigate the attitudes of child and adult psychiatrists towards people with mental health problems. Moreover, this was the first study conducted in Hungary in the field of stigma from the mental health professionals' point of view. Thus, the attitudes of Hungarian adult, child and adolescent psychiatrists and trainees were measured towards their patients with the focus on their link to professional, personal and sociodemographic factors. Twenty-two percent of the practitioners were approached this way who work as a doctor in mental health care in Hungary. The Hungarian psychiatrists' median total scores of the OMS-HC (29 points) were comparable to those of Canadian physicians. The average scores found in the Canadian samples were the following: trainees (n = 35, score = 24.43 points) and specialists in psychiatry (n = 68, score = 22.91 points; and n = 79, score = 30.3 points, respectively) [15, 32]. Similarly to their Canadian colleagues, Hungarian psychiatrists generally hold positive attitudes towards individuals with mental illness since their stigma scores were in the lowest third of the maximum. In Bahrain, the mean stigma scores of mental healthcare professionals were much higher (score = 36.8), suggesting more stigmatizing attitudes than Canadian or Hungarian psychiatrists [14]. Most of the participants (76%) were female, which is in line with the higher proportion and the increasing number of female physicians in Hungary, which is even higher among child psychiatrists. There was no indicated gender ratio specifically for psychiatrists in the Canadian and Bahrainian samples. However, the proportion of females was also higher in the total samples including all professionals: 62% in Bahrain and 77.8% in Canada."

The responses unequivocally demonstrate that the majority of the participating psychiatrists' answers reflect favorable attitudes towards people with mental health problems, and they do not keep much social distance. However, participants had mixed views regarding disclosure and help-seeking. It seems that almost 50% of the sample would be reluctant to seek help for their own mental illness, or at least they would hesitate before doing so and would not disclose the problem to their colleagues. These are in line with the theory that at the workplace, having a mental illness is perceived as less competent, dangerous and unpredictable and that working itself is not considered healthy for people with mental illnesses [33]. When compared to Hungary, a higher proportion of Singaporean medical students were either neutral or reluctant to seek help for their own mental health problems (53.4% vs. 46.9%) and less likely to disclose them to any of their colleagues (86.5% vs. 54.5%) [29]. The results concerning nursing students from Singapore are also higher in the aforementioned areas than in Hungarian psychiatrists. Although the OMS-HC was developed for healthcare workers, there can be differences in the attitudes of different professionals. Our results align with studies that illustrate that psychiatrists or medical students considering becoming a psychiatrist possess more favorable attitudes [9, 13]. Altogether, 16.6% of the respondents agreed that they would see themselves as weak if they had a mental illness and could not fix it themselves. The feeling of being weak due to having a mental illness can also contribute to the obstacles in disclosure, as described in a study on nearly two thousand doctors whose main concerns were being labeled, confidentiality and not understanding the support structures may or may not be in place [34].

Since there is a lack of data on their own mental health problems, it is important to note that information was gathered on the practitioners' lived experience as well. More than half of the respondents (59%) had a close relative who was suffering from a mental illness, nearly half of them (46%) sought help for their own mental health problems at some stage in their life, and one fifth of them (19%) was medically treated for any kind of psychiatric problems.

One of the main findings of the current study is that those who have ever sought help for their own mental problems keep less social distance from their patients and are more likely to present help-seeking behavior in the future as compared to those who have never sought professional help. The above findings are similar to those of Brazilian mental healthcare workers who were psychiatrists in the majority [11]. In Brazil, two-thirds of the sample population had contact with family members suffering from psychiatric disorders, 38% sought help for their own mental health problems, and a quarter of them received a prescription for the problem. They also demonstrated that rare contacts with the affected family member resulted in less stigma towards patients with schizophrenia. However, their own help-seeking behavior was not associated with their stigmatizing attitudes. In parallel with the Brazilians, the stigmatizing attitudes of Asian medical students towards mental illnesses were not related to their own help-seeking behavior either [35].

In contrast, disclosure and help-seeking scores in our study were statistically significantly associated with the following: help-seeking behavior of the participants in the past, the openness to participate in case discussion, supervision or Balint groups and the stigmatizing attitudes of close colleagues. Consequently, those who have more positive experiences with seeking mental help or are ready to talk about their cases and feelings have more favorable attitudes towards their patients. These findings support the aims of stigma reduction programs since workshops promoting conscious engagement with patients with mental illnesses, including group discussion, role-play simulation, debriefing, case discussion and facilitated self-reflection exercises, are able to reach statistically significant decreases in the implicit stigma of participants [36]. Providing psychotherapy for patients, which predicted favorable scores on the attitude subscale in this study, usually also includes case discussion, supervision and skill development. In accordance with the findings in an interventional study, acceptance and

commitment training and multicultural training resulted in a positive impact on stigma among drug abuse counselors. The training in acceptance and commitment also significantly changed the believability of stigmatizing attitudes, and follow-up gained in burnout exceeded those of multicultural training. Clinicians, including psychiatrists, often feel a sense of hopelessness regarding whether they can help people with their skills [37]. The lack of confidence and the perceived lack of competence may reinforce their negative attitudes. Case discussion and Balint groups help professionals to see the person behind their mental illness, reduce anxiety and stress and provide the clinician with new skills, improve awareness to avoid unconscious biases, which are all important factors in lowering stigmatizing attitudes toward patients and mitigating their destructive impact on patients [21, 24, 25].Our findings also draw attention to the important phenomenon when someone's close co-workers stigmatize; then the person likely does so, too. The extent of close psychiatrist colleagues' attitudes, based on the perception of the respondents, was not only associated with but also statistically significantly predicted the attitudes of the respondents towards their patients on the one hand, and the disclosure and help-seeking behavior for their mental health issues on the other. This is consistent with how someone's perceived stigmatizing views of mental illness in other staff members are linked to their self-disclosure. In our sample, 40% of the participants reported that their close colleagues held negative attitudes at least to some extent towards people with mental health problems. A qualitative study from the United Kingdom demonstrated that attitudes of not psychiatrist healthcare professionals towards colleagues with a mental illness were positive; however, they did report that other colleagues held negative attitudes [38]. Lessening the stigma at the workplace among psychiatrists could allow us to intervene since, in work environments, where peer-led initiatives are featured, mental health professionals feel safer when they can disclose their own lived experiences to their colleagues [39].

It should be highlighted that almost half of our study population (43%) had never experienced discrimination for working as a psychiatrist, and this was the strongest predictor of keeping more distance from the patients. The stigma towards psychiatrists [40] and the discipline of psychiatry [41] among healthcare professionals working in different sectors, as well as the general public [42], is an existing issue, which has been reported in many studies. This professional stigma may lead to psychiatrists enacting stigma towards their patients. Associative stigma was linked with depersonalization and emotional exhaustion in a Belgian research study among mental health professionals [18]. Stress and less job satisfaction are fertile soil for more negative attitudes towards both colleagues and patients, and it could have a detrimental effect on patient care [43]. The associative stigma towards psychiatry in Hungary has not been investigated yet; this needs to be addressed.

It should be taken into consideration that investigating stigmatizing views is challenging because the unequivocal demonstration of a positive attitude cannot be disregarded in interviews, especially in self-reported scales. This study aimed to gather information on what psychiatrists think about their attitudes, which could also be biased; however, it is still a less studied area in the broad literature.

Everyone contributes to the stigmatization of people with mental health problems. According to Loch and Rössler, psychiatrists who maintain prejudices are at the intermediate level between the macro level including the general public and mass media, and micro level which includes friends, family, and the individual with mental illness as well via self-stigmatization [44]. Our findings are in line with the literature that psychiatrists generally hold positive views towards patients, but they also prefer to keep a suitable social distance from them. Therefore, even though they have more positive attitudes than other healthcare workers and even though psychiatrists have vast knowledge on mental health, they also contribute negatively to the stigma of the disorder in the inner social circle of the patients. The Royal College of

Psychiatrists recommended that mental health professionals be made aware of their possible prejudices against people with psychiatric problems because such behavior influences the perception of the general public and may contribute negatively to the social acceptance of such people [45].

In accordance with the recommendations, everything that increases awareness and trains the clinician to develop new skills, such as case discussion, psychotherapy could be considered a useful tool in mitigating the stigma in psychiatrists toward patients. Also, since the lived experience with a person with mental health problems is associated with more positive attitudes, education, contact-based education, non-professional personal connection with people with mental illness and skill development are beneficial for psychiatrists.

## Limitations

Our study has some limitations that must be considered to contextualize the reported findings properly. Firstly, the authors tried to contact all Hungarian psychiatrists; however, some were not available e.g. due to long term leave, and it was also difficult to reach colleagues who work in the private sector; therefore, convenience sampling could be considered a potential limitation of the study. It is quite complicated to obtain proper details on the number of practising psychiatrists in Hungary. The response rate was 22% for all psychiatrists (33% for those who work solely in the public health sector) counted based on the estimated number of professionals, which is similar to other studies investigated among psychiatrists (33% and 27.1%) [13, 46]. However, the sample could have been larger to be able to generalize findings for Hungary. Secondly, the authors—not counting the supervisors—are members of the Hungarian Association of Psychiatric Trainees; therefore due to our sampling method and regarding the fact that colleagues were also contacted via social media, approached individuals were mainly young who worked in the public sector. Accordingly, our study population consisted primarily of trainees and young specialists in adult and child psychiatry (54% of the subjects were between 24 and 35 years of age); thus, more experienced practitioners and those who work in private praxis are underrepresented in the study. Therefore, this may limit the generalizability of the results, and comparisons to other studies are limited. On the other hand, the sample represents more the 'next generation' and young colleagues in the field influencing the future. Lastly, considering the number of statistical tests performed, regression results may have limited validity.

## Conclusions

The authors tried to understand better the stigma construct by delving into the stigmatizing attitudes of practising psychiatrists towards their patients. The association was examined between the attitude, disclosure and help-seeking, social distance dimensions of their stigma, and personal and professional factors. The perception of fellow colleagues' attitudes towards patients and the openness to case discussion, supervision and Balint groups were the main two factors that affected the overall attitudes towards patients. These should be considered when tailoring anti-stigma interventions for psychiatrists. In Hungary, the presented data will contribute to a currently ongoing development of a national anti-stigma program.

Although there are many studies on other healthcare providers, only a handful of them investigate the stigmatizing attitudes of psychiatrists. To the best of our knowledge, this is the only comprehensive study conducted exclusively on psychiatrists that draws a link between the stigma towards patients and the presence of psychotherapeutic activity, interest in case discussion groups and the stigma towards psychiatry. According to our results, the favorable attitudes of psychiatrists are associated with their own lived experiences with any kind of psychiatric condition along with previous help-seeking behavior and the opportunity to work

together with fellow psychiatrists, whose attitudes are less stigmatizing. The Hungarian psychiatrist population generally holds positive attitudes towards their patients; however, this study highlights the need to develop anti-stigma programs for psychiatrists. In the future, multi-center studies on psychiatrists' attitudes could provide essential insights into the differences between countries since there is a lack of international studies investigating the stigma towards patients. On the other hand, longitudinal studies examining factors associated with stigma would expand our existing knowledge.

## Supporting information

**S1 Questionnaire. Questionnaire in English including the sociodemographic and professional questions and the Opening Minds Stigma Scale.**
(DOCX)

## Acknowledgments

We are thankful to the Hungarian Psychiatric Association for helping us in the dissemination of the survey. We are sincerely grateful to András Mihály Boros MD PhD (Heart and Vascular Center, Semmelweis University, Budapest, Hungary) for language editing and proofreading, to Attila Pulay MD PhD (Department of Psychiatry and Psychotherapy, Semmelweis University, Budapest, Hungary) for his expert comments on the statistical analysis, and to, Lili Fókás for professional data visualization. Finally, we would like to express our gratitude to every Hungarian psychiatrist colleague who participated in our study.

## Author Contributions

**Conceptualization:** Dorottya Őri, Péter Szocsics, Tamás Molnár, Fanni Virág Ralovich, Ágnes Bene, György Purebl.

**Data curation:** Dorottya Őri.

**Formal analysis:** Dorottya Őri, Sándor Rózsa.

**Funding acquisition:** Ágnes Bene.

**Investigation:** Dorottya Őri.

**Methodology:** Dorottya Őri, Péter Szocsics, Tamás Molnár, Zsolt Huszár, Sándor Rózsa, György Purebl.

**Project administration:** Dorottya Őri, Péter Szocsics, Tamás Molnár, Fanni Virág Ralovich, Zsolt Huszár, Ágnes Bene, György Purebl.

**Supervision:** Zsuzsa Győrffy, György Purebl.

**Validation:** Sándor Rózsa.

**Visualization:** Dorottya Őri.

**Writing – original draft:** Dorottya Őri.

**Writing – review & editing:** Dorottya Őri, Péter Szocsics, Tamás Molnár, Fanni Virág Ralovich, Zsolt Huszár, Ágnes Bene, György Purebl.

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
