## [Decision Letter · Decision Letter 0]

18 Apr 2022

PONE-D-21-25580Stigma towards mental illness and help-seeking behaviors among adult and child psychiatrists in Hungary: a cross-sectional studyPLOS ONE

Dear Dr. Györffy,

Thank you for submitting your manuscript to PLOS ONE. After careful consideration, we feel that it has merit but does not fully meet PLOS ONE’s publication criteria as it currently stands. Therefore, we invite you to submit a revised version of the manuscript that addresses the points raised during the review process.

We look forward to receiving your revised manuscript.

Kind regards,

Stephan Doering, M.D.

Academic Editor

PLOS ONE

Journal Requirements:

Reviewers' comments:

Reviewer's Responses to Questions

**Comments to the Author**

1. Is the manuscript technically sound, and do the data support the conclusions?

Reviewer #1: Yes

Reviewer #2: Yes

2. Has the statistical analysis been performed appropriately and rigorously? 

Reviewer #1: Yes

Reviewer #2: I Don't Know

3. Have the authors made all data underlying the findings in their manuscript fully available?

Reviewer #1: No

Reviewer #2: Yes

4. Is the manuscript presented in an intelligible fashion and written in standard English?

Reviewer #1: No

Reviewer #2: Yes

5. Review Comments to the Author

Reviewer #1: This was an excellent paper with a relativity large sample size and encouraging to find some interventions to help combat stigma. It is clear alot of work has been put into this. The main critique of the paper comes from some issues with the English wording and use of 1st person (we, he) rather than 3rd person tense.). There was some informal language also “nearly every other psychiatrist” and acknowledging “ my best friend” which should be removed. I would recommend having an English proof reader look over the paper before it is published.

It was also be useful to know what questions were asked of the participants- maybe this could be in the appendix. It would also be good to comment that most of the participants were female.

I’m also interested to know more about the evidence of using ballint groups and link to psychotherapy as indications of people having less stigmatising views. This would explain why you chose to ask these particular questions. I think this should be included in the introduction and some of the other studies which are less relevant could be cut out. In particular, the conflicting studies about how stigmatising a group psychiatrists are in general, could be summarised to say that there are conflicting evidence about psychiatrists holding stigmatising views. More detail about the large Canadian study which this study seems to have compared itself to would also be useful in the introduction.

To summarise, this was a great paper but needs a bit of refinement before being published. There is a good opportunity to cut back on the word count from the introduction and focus on studies which are more relevant to the study. It would also be worthwhile having an professional English proofreader review the study so that it reads less informally.

Reviewer #2: Stigma towards mental illness and help seeking behaviours amount adult and child psychiatrist in Hungary: across sectional study

Thank you for the opportunity to review. Overall this is an interesting paper with new information to add to the field. I have the following questions and comments/suggestions for improvement for the authors’ consideration:

1. The notion of stigma is complex and the semantics can be confusing. Suggest the authors elaborate more clearly what they mean by stigma in a number of places throughout the Introduction. For example, page 4 first paragraph last sentence: “… the support of professionals might play and essential role in mitigating stigma since higher levels of burnout were associated with more negative attitudes towards patients”. It is unclear what are the authors mean - support for professionals will mitigate having stigmatizing attitudes, or mitigating stigma in their patients, or in society in general. Page 3, last sentence -…”demand for personal accomplishment and burn out or predictors of stigma…” Again this stigma is unclear in what the authors are referring to. Please review carefully throughout the introduction.

2. Page 3, it will be helpful to explain the Weak-not-sick scale for the readership.

3. A broader review of how to support mental health workers will be helpful, rather than providing two choices such as case discussion or a Balint group. It’s as if these are already well accepted and only choices. If there are evidence and literature to support their effectiveness please provide such.

4. Page 4, Methods. The research group contacted 60 child psychiatric inpatient services and 69 outpatient services with many other efforts. It will be helpful to estimate what are the total number of potential recipients here (only later mentioned in the Results ), and the percentage that the 238 respondents represent. The response rates seem very low, and more analysis and explanations of the rate of response would be helpful. This should be a limitation in the discussion as well.

5. Page 4, last paragraph that talks about the Hungarian version of the OMSHC seem out of place in describing the process of translation and concept checking, after citing the psychometric properties and the test retest reliability of the Hungarian version already in the preceding paragraph. This could be either incorporated in introducing this Hungarian version or be omitted.

6. Also it is unclear what do the authors mean by “a good concurrent validity was calculated with a mental illness clinicians attitude- 4 scale. More explanation is needed.

7. Under statistics, there is no mention of multi test corrections since this is a large item by item analyses in correlations and regressions. A statistics expert reviewer may be needed.

8. There are a number of small grammatical errors that could be checked. For examples: Page 14 second last paragraph “practitioner’s personal affection as well” is poor choice of words. Missing a verb in the sentence about nursing students from Singapore, and they were “one fifth” on top of page 15, etc.

9. In the Discussion I wonder whether the common phenomenon of people wanting to provide socially approved answers, particularly given that the survey is for medical professionals should be considered as a factor for discussion. The authors do talk confidently about “unequivocal demonstration of a positive attitude.” This is a worthwhile in any survey to consider, even if done anonymously.

10. One would hope for more in-depth reflections and discussions on why certain findings are the way they are, in addition to the numerical comparisons with other settings such as those in Singapore or Brazil or Bahrain. These comparisons themselves do not give rise to understanding beyond a cross cultural difference that the authors are bringing to light. The explanations about why a group discussion helps to lower stigma is on the right track, but it could be further illuminated on why such contact or discussion would lead to lowering of stigma.

11. Are there theories within mental health stigma that the authors could invoke.? It would be helpful to find a theory that the authors would like to anchor some of their findings rather than relying on statistical correlational results as an explanation in and of itself.

12. Last paragraph, page 16, interesting to highlight that psychiatrist who have experienced discrimination have more distance from their patients. Do the authors have some understanding in theory rather than stating that it’s an existing issue that needs to be further addressed?

13. Limitations: the very low response rate, the statistical corrections needed, why were some people not available, what are the reasons behind her low response and why is the sampling method more likely to reach and get responses from the younger individuals, what biases might have these selective samples introduced to the results should all be explored and discussed.

14. Strength of the paper is, as the authors also concluded, that there is a limited number of studies on psychiatrists and issues related to stigma. And it’s a solid study that can be further improved.

6. PLOS authors have the option to publish the peer review history of their article (what does this mean?). If published, this will include your full peer review and any attached files.

Reviewer #1: No

Reviewer #2: No

---

## [Author Response · Author response to Decision Letter 0]

29 May 2022

Journal Requirements:

Thank you very much for the comments, we corrected the manuscript accordingly and uploaded the dataset to a public repository.

Reviewer #1

This was an excellent paper with a relativity large sample size and encouraging to find some interventions to help combat stigma. It is clear a lot of work has been put into this. The main critique of the paper comes from some issues with the English wording and use of 1st person (we, he) rather than 3rd person tense.). There was some informal language also “nearly every other psychiatrist” and acknowledging “my best friend” which should be removed. English proof reader look over the paper before it is published.

It was also be useful to know what questions were asked of the participants- maybe this could be in the appendix. It would also be good to comment that most of the participants were female.

I’m also interested to know more about the evidence of using Balint groups and link to psychotherapy as indications of people having less stigmatising views. This would explain why you chose to ask these particular questions. I think this should be included in the introduction nd some of the other studies which are less relevant could be cut out. In particular, the conflicting studies about how stigmatising a group psychiatrists are in general, could be summarised to say that there are conflicting evidence about psychiatrists holding stigmatising views. More detail about the large Canadian study which this study seems to have compared itself to would also be useful in the introduction.

To summarise, this was a great paper but needs a bit of refinement before being published. There is a good opportunity to cut back on the word count from the introduction and focus on studies which are more relevant to the study. It would also be worthwhile having an professional English proofreader review the study so that it reads less informally.

Thank you very much for taking the time to review our work and provide us with your expertise. Your suggestions are highly appreciated; we have taken into consideration all of the comments and recommendations you raised.

Based on your suggestion, we sent our manuscript to an English proofreading service, who thoroughly corrected the English grammar and wording. We also changed the informal language to more formal wording to maintain a consistent register throughout.

We have enclosed the questionnaire as supplementary material.

In Hungary, the majority of psychiatrists are female, especially child psychiatrists. Since child psychiatrists were overrepresented in the sample, following your insightful suggestion, we added the following to the discussion to comment.

“Most of the participants (76%) were female, which is in line with the higher proportion and the increasing number of female physicians in Hungary, which is even higher among child psychiatrists. There was no indicated gender ratio specifically for psychiatrists in the Canadian and Bahrainian samples. However, the proportion of females was also higher in the total samples including all professionals: 62% in Bahrain and 77.8% in Canada.”

It is worth noting the Canadian study included a diverse sample of healthcare professionals and the Bahrainian study consisted of mental healthcare workers.

We have conducted an extensive literature search, and have added additional pertinent information concerning the positive effect of case discussion, psychotherapy, and Balint groups on the attitudes of psychiatrists towards patients. In fact, the idea of Balint groups was derived from the lived experience of the authors as well. We are all active Balint group attendees and feel these sessions are very helpful in the opinion-forming of our patients and we generally get a better understanding of the patient-doctor relationship during these sessions and feel more empathy towards them.

I am additionally grateful to you for the suggestion of cutting back the word count. We abbreviated the general introduction, and summarized the conflicting studies in a sentence based on your comment, and added more detail about the Canadian study. However, the other reviewer requested some additional information in the introduction, therefore, the overall manuscript is not shorter than it was before.

We are grateful to you for drawing our attention to all of these issues. Again, thank you very much for taking the time to review our manuscript and for the important suggestions.

Reviewer #2

Stigma towards mental illness and help seeking behaviours amount adult and child psychiatrist in Hungary: across sectional study

Thank you for the opportunity to review. Overall this is an interesting paper with new information to add to the field. I have the following questions and comments/suggestions for improvement for the authors’ consideration:

Thank you very much for taking the time to review our work and provide us with your expertise. Your suggestions are highly appreciated; we have taken into consideration all of the comments and recommendations you raised.

1. The notion of stigma is complex and the semantics can be confusing. Suggest the authors elaborate more clearly what they mean by stigma in a number of places throughout the Introduction. For example, page 4 first paragraph last sentence: “… the support of professionals might play an essential role in mitigating stigma since higher levels of burnout were associated with more negative attitudes towards patients”. It is unclear what are the authors mean - support for professionals will mitigate having stigmatizing attitudes, or mitigating stigma in their patients, or in society in general. Page 3, last sentence -…”demand for personal accomplishment and burnout or predictors of stigma…” Again this stigma is unclear in what the authors are referring to. Please review carefully throughout the introduction.

Thank you for drawing our attention to the issue that the introduction should be improved in clarity. We identified several parts in the introduction that lacked clarity.

2. Page 3, it will be helpful to explain the Weak-not-sick scale for the readership.

Thank you, we would have been happy to give more details about the Weak-not-sick scale. However, based on the other reviewer’s request we tried to shorten the introduction -in particular the part concerning the conflicting studies about how stigmatising a group of psychiatrists are in general. Therefore, we cited the research paper, but omitted the text about the Weak-not-sick scale. Thank you for your understanding.

3. A broader review of how to support mental health workers will be helpful, rather than providing two choices such as case discussion or a Balint group. It’s as if these are already well accepted and only choices. If there are evidence and literature to support their effectiveness please provide such.

Thank you very much for your suggestion. We listed other approaches rather than case discussion or Balint groups that support clinicians. Please note that an extensive literature review could not have been done because the other reviewer requested the shortening of the introduction and listing all the current available means of support for mental health workers would go beyond this manuscript.

We provided more evidence both on the possible support of psychiatrists linked to our findings with the main focus on Balint- and case discussion groups in the introduction, and the discussion sessions based on your suggestion.

4. Page 4, Methods. The research group contacted 60 child psychiatric inpatient services and 69 outpatient services with many other efforts. It will be helpful to estimate what are the total number of potential recipients here (only later mentioned in the Results ), and the percentage that the 238 respondents represent. The response rates seem very low, and more analysis and explanations of the rate of response would be helpful. This should be a limitation in the discussion as well.

Thank you very much for raising this crucial issue. We have added more details to the methods section. There is no proper data on the number of psychiatrists who work in Hungary, however, based on the estimation after contacting each medical university and the professional associations we learned that altogether in the public sector a maximum of 635 practitioners work in the field of general adult and child psychiatry (including trainees). It is quite difficult to discern the exact numbers since some colleagues work only a few hours a week, or are passive, or are on leave from work for a longer period. Furthermore, they were also counted as full-time workers.

We contacted all psychiatric services in Hungary, the majority of the outpatient services are in the countryside, in which only 1 or 2 psychiatrists work -especially in the field of child psychiatry. (Telling the truth, it is a very sad fact that there are some counties where there is only 1 specialist for 2 counties).

The response rate was 238/635= 0.3748. Physicians are often a group with low survey response rates and we have looked for research that investigates the response rates among physician specialists. According to a Canadian study done in 2015, which aimed to explore the response rates of specialists, 27.1% of psychiatrists responded (Cunningham et al. 2015). An Australian study’s rate was quite similar to ours (33% for psychiatrists) (Reavley et al. 2013).

We contacted the head of the service in each case, it might have been the case that he or she did not forward our survey to their colleagues.

We were not able to reach those colleagues who work in the private sector, this was also mentioned in the limitation section.

To sum up, we have included the more detailed numbers in the methods section and added relevant information in the limitations. Thank you for raising the question on the numbers!

5. Page 4, last paragraph that talks about the Hungarian version of the OMSHC seem out of place in describing the process of translation and concept checking, after citing the psychometric properties and the test retest reliability of the Hungarian version already in the preceding paragraph. This could be either incorporated in introducing this Hungarian version or be omitted.

Thank you very much for your suggestion, we share your views and see the unnecessity of the last paragraph, therefore, we have omitted this part.

6. Also it is unclear what do the authors mean by “a good concurrent validity was calculated with a mental illness clinicians attitude- 4 scale. More explanation is needed.

Thank you very much for requesting more detail. This was also described in our previous article on the psychometric properties of the scale. Therefore, we have added the intraclass correlation coefficient (value is considered good above 0.7) and its 95% confidence interval to the sentence to give more details about the concurrent validity.

“Its test-retest reliability was found to be excellent for the total score and good for the three subscales, and a good concurrent validity was measured for the total score with the Mental Illness: Clinician’s Attitudes-4 scale (intraclass correlation coefficient value was 0.77 [95% confidence interval ranged between 0.11 and 0.92]).„ 

7. Under statistics, there is no mention of multi test corrections since this is a large item by item analyses in correlations and regressions. A statistics expert reviewer may be needed.

Thank you very much for your valuable comment. The number of variables, and the relatively low number of cases could be important limitations. In the case of non-parametric ANOVA -where more than two groups were analysed, for example, ‘Close psychiatrist colleagues’ stigmatizing attitudes towards their patients’, the multiple comparison analyses could have been calculated with Bonferroni correction. However, we did not perform multiple comparisons between groups, therefore multi-test correlations were not appropriate.

In the case of linear regression, multi-correlation tests are not built into the SPSS software package. We used instead the bootstrap method to construct 95% confidence intervals of the results to be 95% confident the interval contains the true value of beta estimates.

Therefore, based on your comment, we have included the following sentence in the limitations section:

Lastly, considering the number of statistical tests performed, regression results may have limited validity.

8. There are a number of small grammatical errors that could be checked. For examples: Page 14 second last paragraph “practitioner’s personal affection as well” is poor choice of words. Missing a verb in the sentence about nursing students from Singapore, and they were “one fifth” on top of page 15, etc.

Thank you very much for drawing attention to the grammatical errors. Since both the reviewers highlighted the English language issues, we decided to send the manuscript to a professional proofreader service before the resubmission. We also changed the “practitioner’s personal affection as well” to the following: “lived experience” and corrected the issue of the missing word as well.

9. In the Discussion I wonder whether the common phenomenon of people wanting to provide socially approved answers, particularly given that the survey is for medical professionals should be considered as a factor for discussion. The authors do talk confidently about “unequivocal demonstration of a positive attitude.” This is a worthwhile in any survey to consider, even if done anonymously.

Thank you very much for the important comment. We totally agree with the necessity to discuss this phenomenon. Based on your suggestion, we have added the following to the discussion section to raise awareness of this:

“It should be taken into consideration that investigating stigmatizing views is challenging because the unequivocal demonstration of a positive attitude cannot be disregarded in interviews, especially in self-reported scales. This study aimed to gather information on what psychiatrists think about their attitudes, which could also be biased; however, it is still a less studied area in the broad literature.”

10. One would hope for more in-depth reflections and discussions on why certain findings are the way they are, in addition to the numerical comparisons with other settings such as those in Singapore or Brazil or Bahrain. These comparisons themselves do not give rise to understanding beyond a cross cultural difference that the authors are bringing to light. The explanations about why a group discussion helps to lower stigma is on the right track, but it could be further illuminated on why such contact or discussion would lead to lowering of stigma.

Thank you very much for this excellent comment.

The significant difference between the Hungarian psychiatrists and the Bahreinian mental healthcare workers might appear to stem from the fact that mental health is not prioritized and the resources are scarce in Arabic countries, and people with mental illness experience the compounded disadvantages of poverty and illness stigma according to a study on Arabic families (Dardas et al. 2015). It is also more complicated to discuss because as written that the Bahrainian was a diverse sample of mental healthcare workers.

In Asian cultures like Singapore, mental health stigma might be influenced by traditional beliefs and mindsets, and honor and collectivism are valued highly. Higher levels of collectivism are associated with higher levels of stigma (Papadopoulos et al., 2013). People from Southern Asia seem to fear negative social consequences of disclosure and believe that sharing negative information would cause a burden to others (Chen et al., 2013).

After discussing the cultural differences with the whole research team, we agreed that the discussion of culture is a very important area. However, it cannot be properly done without the involvement of a researcher from the given country to avoid any biases or misunderstanding, since only local researchers can give us a valuable insight in their culture. Therefore, we prefer not to include additional information on the cultural differences.

However, we have added more information about case discussion and Balint groups in both the introduction and the discussion sections based on your important suggestion.

11. Are there theories within mental health stigma that the authors could invoke.? It would be helpful to find a theory that the authors would like to anchor some of their findings rather than relying on statistical correlational results as an explanation in and of itself.

Thank you very much for the suggestion. We have added the text below to contextualize our findings in the stigma theory. Thank you again, this definitely improved the discussion and highlighted the take-home message.

“Everyone contributes to the stigmatization of people with mental health problems. According to Loch and Rössler, psychiatrists who maintain prejudices are at the intermediate level between the macro levelincluding the general public and mass media, and micro level which includes friends, family, and the individual with mental illness as well via self-stigmatization (Gaebel et al., 2017). Our findings are in line with the literature that psychiatrists generally hold positive views towards patients, but they also prefer to keep a suitable social distance from them. Therefore, even though they have more positive attitudes than other healthcare workers and even though psychiatrists have vast knowledge on mental health, they also contribute negatively to the stigma of the disorder in the inner social circle of the patients. The Royal College of Psychiatrists recommended that mental health professionals be made aware of their possible prejudices against people with psychiatric problems because such behavior influences the perception of the general public and may contribute negatively to the social acceptance of such people (Crisp et al., 2004).

 In accordance with the recommendations, everything that increases awareness and trains the clinician to develop new skills, such as case discussion, psychotherapy could be considered a useful tool in mitigating the stigma in psychiatrists toward patients. Also, since the lived experience with a person with mental health problems is associated with more positive attitudes, education, contact-based education, non-professional personal connection with people with mental illness and skill development are beneficial for psychiatrists.”

12. Last paragraph, page 16, interesting to highlight that psychiatrists who have experienced discrimination have more distance from their patients. Do the authors have some understanding in theory rather than stating that it’s an existing issue that needs to be further addressed?

Thank you so much for raising this question. We have added more detail on this topic in the discussion section mainly focusing on job satisfaction and the exhaustion of clinicians, which result in more stigma toward patients.

“This professional stigma may lead to psychiatrists enacting stigma towards their patients. Associative stigma was linked with depersonalization and emotional exhaustion in a Belgian research study among mental health professionals (Verhaeghe et al., 2012). Stress, less job satisfaction are fertile soil for more negative attitudes towards both colleagues and patients, and it could have a detrimental effect to patient care.”

13. Limitations: the very low response rate, the statistical corrections needed, why were some people not available, what are the reasons behind her low response and why is the sampling method more likely to reach and get responses from the younger individuals, what biases might have these selective samples introduced to the results should all be explored and discussed.

Thank you for requesting additional information for this section, being aware of the limitations is crucial and the readers should be informed appropriately. We have completed the results section with the following:

“It is quite complicated to obtain proper details on the number of practising psychiatrists in Hungary. The response rate was 22% for all psychiatrists (33% for those who work solely in the public health sector) counted based on the estimated number of professionals, which is similar to other studies investigated among psychiatrists (33% and 27.1%) (reference provided). However, the sample could have been larger to be able to generalize findings for Hungary. Secondly, the authors - not counting the supervisors - are members of the Hungarian Association of Psychiatric Trainees; therefore due to our sampling method and regarding the fact that we also contacted colleagues by using social media, we approached mainly young individuals who worked in the public sector. Accordingly, our study population consisted primarily of trainees and young specialists in adult and child psychiatry (54% of the subjects were between 24 and 35 years of age); thus, more experienced practitioners and those who work in private praxis are underrepresented in the study. Therefore, this may limit the generalizability of the results and comparisons to other studies are limited. On the other hand, the sample represents more the ‘next generation’ and young colleagues in the field influencing the future.”

14. Strength of the paper is, as the authors also concluded, that there is a limited number of studies on psychiatrists and issues related to stigma. And it’s a solid study that can be further improved.

Thank you so much for your valuable review of our manuscript, and for providing us with your expertise. Your suggestions are highly appreciated; we have taken into consideration all of the comments and recommendations you raised, they have definitely improved the manuscript and have helped us to provide a much more comprehensive picture to future readers.

---

## [Editor Report · Decision Letter 1]

31 May 2022

Stigma towards mental illness and help-seeking behaviors among adult and child psychiatrists in Hungary: a cross-sectional study

PONE-D-21-25580R1

Dear Dr. Ori,

We’re pleased to inform you that your manuscript has been judged scientifically suitable for publication and will be formally accepted for publication once it meets all outstanding technical requirements.

Kind regards,

Stephan Doering, M.D.

Academic Editor

PLOS ONE

---

## [Editor Report · Acceptance letter]

2 Jun 2022

PONE-D-21-25580R1 

Stigma towards mental illness and help-seeking behaviors among adult and child psychiatrists in Hungary: a cross-sectional study 

Dear Dr. Őri:

I'm pleased to inform you that your manuscript has been deemed suitable for publication in PLOS ONE. Congratulations! Your manuscript is now with our production department. 

Kind regards, 

on behalf of

Professor Stephan Doering 

Academic Editor

PLOS ONE